# Woody species diversity, structure and community distribution along environmental gradients of Seqela Dry Afromontane forest in Northwestern Ethiopia

Liyew Birhanu[1,2]*, Getaneh Moges[1], Nigussie Amsalu[1], Heiko Balzter[2,3]

1 Department of Biology, Debre Markos University, Debre Markos, Ethiopia, 2 School of Geography, Geology and the Environment, Institute for Environmental Futures, University of Leicester, Leicester, United Kingdom, 3 National Centre for Earth Observation, University of Leicester, Leicester, United Kingdom

* liyewmtu@gmail.com, liyew_birhanu@dmu.edu.et

**Data Availability Statement:** All relevant data are within the article and its supporting information files.

## Abstract

Dry evergreen Afromontane forests are severely threatened due to the expansion of agriculture and overgrazing by livestock. The objective of this study was to investigate the composition of woody species, structure, regeneration status and plant communities in Seqela forest, as well as the relationship between plant community types and environmental variables. Systematic sampling was used to collect vegetation and environmental data from 52 (20 m x 20 m) (400 m2) plots. Density, Diameter at Breast Height (DBH), basal area, frequency, and importance value index (IVI) of woody species were computed to characterize the vegetation structure of the forest. Agglomerative hierarchical cluster analysis and Canonical Correspondence Analysis (CCA) with R software were used to identify plant communities and analyse the relationship between plant community types and environmental variables, respectively. A total of 68 woody plant species belonging to 63 genera and 44 families were identified. The Shannon diversity index and evenness values of the study area were 2.12 and 0.92, respectively. The total basal area and density of woody species were 27.4 m$^2$ ha$^{-1}$ and 1079.3 individual ha$^{-1}$, respectively. The most frequent woody species in the Seqela forest included *Albizia gummifera* (51.92%), *Croton macrostachyus* (44.23%), *Olinia rochetiana* and *Teclea nobilis* (36.54%). Additionally, the most dominant species, as indicated by their importance value index (IVI), were *Erythrina brucei* (IVI = 11.24), *Prunus africana* (IVI=8.68), and *Croton macrostachyus* (IVI=7.38). Four plant community types were identified: *Albizia gummifera - Ekebergia capensis*, *Prunus africana - Croton macrostachyus*, *Vachellia abyssinica - Dombeya torrida* and *Schefflera abyssinica - Teclea nobilis*. The CCA results showed that the variation of species distribution and plant community formation were significantly (P < 0.05) related to altitude, organic matter, aspect, slope and soil available phosphorus. The regeneration status assessment of the forest revealed a good regeneration status, which was linked to diverse and abundant seed bank in the soil can ensure a continuous supply of seeds for regeneration; therefore, it is recommended to implement periodic soil seed bank assessments to monitor seed diversity and abundance and inform targeted conservation actions.

**Funding:** This research did not receive any specific grants from funding agencies. H.B. received financial support from the Natural Environment Research Council (NERC) via the National Centre for Earth Observation (NCEO).

**Competing interests:** The authors declare that they have no competing interests.

## 1. Introduction

Forest ecosystems are important repositories of terrestrial biodiversity and play a key role in influencing socio-ecological and cultural attributes of human societies including livelihood activities of traditional societies living as well as associated to these forests [1–3]. These ecosystems are highly threatened by anthropogenic activities [4–6]. Poverty, population pressure, agricultural expansion/intensification and infrastructural development are the major threats to biodiversity and environment [7–9]. Altitude and latitude have gradients associated to climatic parameters [10–12] which influence the species distribution and their function [13, 14]. The fragmentation of a population at an environmental gradient due to disturbance (agents that change the population structure and dynamics), be it anthropogenic or natural, may result in different population growth and vital rates among the forest ecosystems [15, 16].

Ethiopia's forests are essential natural resources that offer significant socioeconomic and cultural benefits to local communities [17]. Moreover, these forests play a crucial role in providing various ecosystem services, such as biodiversity conservation, carbon sequestration, and watershed management [18]. However, despite their vital importance, the forest cover in Ethiopia has faced significant threats due to rapid population growth and land use change [19]. According to a recent study [20], Ethiopia's forest cover has decreased by 16% between 1990 and 2020, despite the revised definition of forests that includes woodlands. Deforestation continues to be a major concern in the country. The diverse environmental conditions in Ethiopia, including variations in altitude, climate, and soil, as well as human interactions, result in varying plant community distribution, structure, and composition [21].

On regional and global scales, climatic factors shape plant species responses. In contrast, at local levels, topographic and soil (edaphic) conditions play a crucial role in the formation and distribution of plant communities [22]. Altitude is a key topographic variable that affects plant species distribution and community formation by altering atmospheric pressure, humidity, temperature, soil type, and microclimatic conditions. Slope also impacts plant distribution by affecting soil nutrient dynamics [22–24]. Additionally, soil properties and disturbances influence plant species distribution and community structure. Understanding these complex interactions is crucial for effective conservation and management of forest ecosystems, especially in regions like Ethiopia, where diverse environmental conditions [24].

The dry evergreen Afromontane forests in Ethiopia, including the Seqela Dry Afromontane forest, are particularly threatened by agricultural expansion, overgrazing, and population growth [17, 19]. Understanding the distribution patterns of plant species and the environmental factors influencing them is crucial for effective biodiversity conservation [25, 26]. While several studies have examined floristic composition and distribution patterns in dry evergreen Afromontane Forests [22, 27–30], there is a lack of specific research on the Seqela Dry Afromontane Forest. Thus, this study was designed to address the following objectives: (1) to document the woody species composition of Seqela Forest; (2) to assess the vegetation structure and regeneration status of Seqela Forest; (3) to identify plant community types in the study area; and (4) to identify the major environmental factors determining species distribution and community composition in Seqela Forest.

## 2. Materials and methods

### 2.1 Description of the study area

The study was carried out in the Seqela Forest, which is located in Northwestern Ethiopia's Quarit District of the West Gojjam Zone of the Amhara Regional State (Fig 1). The district is located 418 kilometers northwest of Addis Ababa and 233 kilometers southeast of Bahir Dar

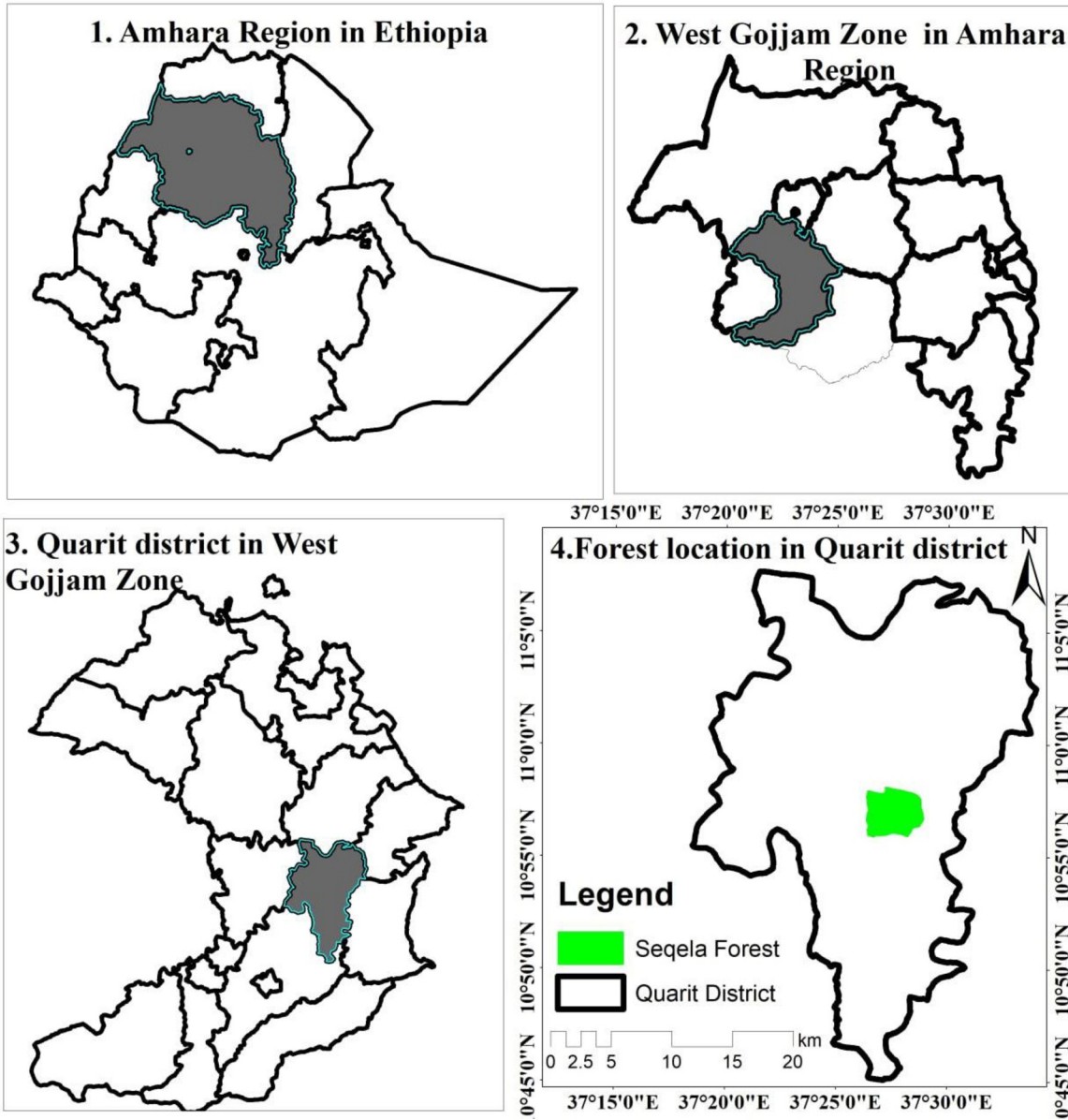

**Fig 1. Seqela forest located in Quarit district of West Gojjam Zone, Amhara Region in Ethiopia (Our study does not need to supply a copy right notice for Fig 1.** Because the location map of the study area (Fig 1) shape file data was obtained from Ethiopian Mapping Agency (https://africaopendata.org/dataset/ethiopia-shapefiles) is free and open to researchers.

city. Seqela forest is categorised as dry evergreen Afromontane forest [31]. The forest is located between 100°56'0" and 100°57'30" N and 370°28'0" and 370°28'30" E. The total size of the forest is 197.6 hectares. Eutricvertsols, a subset of vertisols, are the most common form of soil in the research area [32]. The 10-year climate data (2012-2021) for the Quarit station was obtained from the National Meteorological Services Agency. The average annual rainfall in the district is 1393 mm. The average mean minimum and maximum temperatures in the study area are 10°C and 30.70°C, respectively, with an average annual temperature of about 20.10°C (Fig 2). The main crop types in the district include cereal crops, oil crops, pulses, and vegetables. Cereal crops include barley (*Hordeum vulgare*), wheat (*Triticum sp.)*, teff (*Eragrostis tef*),

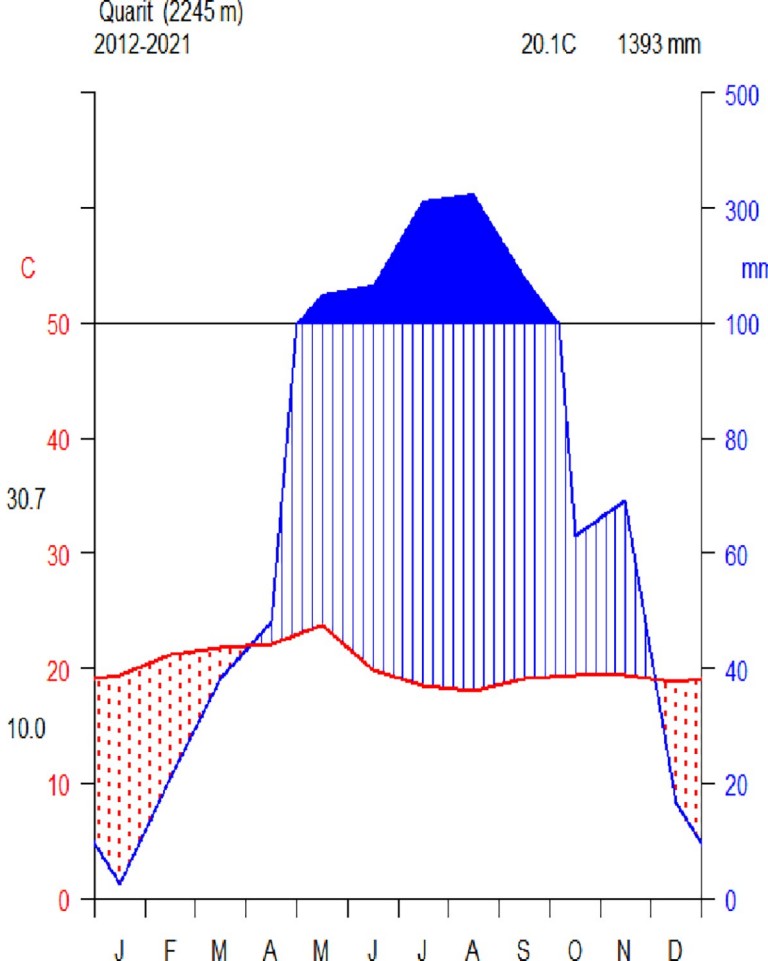

**Fig 2. Climate diagram of Quarit (data source: NMSA, 2022).**

and maize (*Zea mays*). Oil crops include Niger seed (*Guizotia abyssinica*) and sunflower (*Helianthus annus*). Pulses include faba bean (*Vicia faba*) and field pea (*Pisum sativum*). Vegetables include cabbage (*Brassica oleracea*), potatoes (*Solanum tuberosum*), garlic (*Allium sativum*), and onion (*Allium cepa*). Potatoes are the primary traded tuber crop [33]. Letters of consent were taken from Debre Markos University and Quarit District Agriculture and Rural Development Offices before data collection. Thus, we obtained permission to collect both vegetation and environmental data and conduct our research in the study area.

## 2.2 Sampling design

A reconnaissance study was conducted across the forest to determine the sampling plots for the collection of environmental and vegetation data, as well as to get an overview of the site characteristics. To collect data on vegetation and the environment, a systematic sampling method was applied. A total of 52 plots, 20 m x 20 m along transects following altitudinal gradient were laid down. The distance between two consecutive plots and transect lines was 100 and 200 m apart in the forest, respectively. Five smaller subplots of 5 x 5 m$^2$, four at the corner, and one at the center of the main plot, were established for seedling and sapling species data collection [34].

## 2.3 Vegetation data collection

Trees and shrubs larger than 2.5 cm DBH were counted individually in each plot and all woody plant species recorded. For shrubs and trees larger than 2.5 cm in diameter, the diameter at breast height (DBH) was measured and recorded for plants with DBH >2.5 cm. Plants taller than 2.5 m were considered mature trees and shrubs, whereas saplings between 1 and 2.5 m and seedlings up to 1 m tall [35]. All collected plant specimens were pressed, dried and taken to the Herbarium of Debre Markos University, for taxonomic identification using the published books of Flora of Ethiopia and Eritrea. Finally, the presence-absence and cover abundance data were converted to the modified 1–9 Braun Blanquet scale [36].

## 2.4 Environmental data collection

For every plot within the forest, environmental data such as altitude, longitude and latitude were captured using GPS. Each sample plot's slope was determined using a Suunto clinometer. The elevation of the study forest ranges from 2314 to 2577 m above sea level. On the other hand, the slope classes range from 12.41 to 36.87 Degrees. As a potential indicator of total solar energy, the aspect was coded based on [37] as follows: N = 0; E = 2; S = 4; W = 2.5; NE = 1; SE = 3; SW = 3.3; and NW = 1.3. Five locations, each sized 15 × 15 cm and with a depth of 0–30 cm, were used to gather soil samples from each main plot (four from each corner and one from the center) [38]. After combining these samples to create a composite sample, about 1 kg was used for examination/analysis. The soil analysis was assessed at Debre Markos Soil Research and Fertility Improvement's soil laboratory. Before chemical tests, the soil samples were air dried, crushed, and sieved with a 2 mm screen size to remove organic waste and root particles. The soil samples were analyzed for pH, electrical conductivity (EC), Cation exchange capacity (CEC), soil organic matter (SOM), total nitrogen (N), available phosphorus (P), and soil texture (sand, silt, and clay), following the standard procedures described in [39]. Soil pH and EC were measured using a glass electrode pH-meter with a 1:2.5 soil-to-water ratio. Organic carbon content was determined using the Walkley-Black method, and organic matter was calculated by multiplying the organic carbon content by 1.724 (OM = 1.724 x OC). Total nitrogen was measured through the Kjeldahl method after digestion with concentrated sulfuric acid. Available phosphorus was was determined by the UV/vis spectrophotometer, following the Olson method. Cation exchange capacity (CEC) was determined after extracting the soil samples by ammonium acetate (1N NH4OAc) at pH 7.0.

## 2.5 Data analysis

**2.5.1 Diversity and similarity indices.** Shannon-Wiener diversity index and Shannon's evenness were computed to describe the species diversity of the study area [40].

H'= -$\sum_{i=1}^{s} pi \ln pi$, where by

H' = Shannon diversity index

**s** = number of species,

p= proportion of individuals or abundance of the ith species expressed as a proportion of total cover in the sample; and *ln* = natural logarithm.

Shannon's evenness index (J) was also calculated using: J = $\frac{H'}{H_{max}} = \frac{-\sum_{i}^{s} pi\ lnpi}{ln\ s}$

where H' = Shannon–Wiener Diversity Index; and H'max = lns, where s is the number of species in the sample.

**2.5.2 Species similarity analysis.** So as to compare species composition similarity among community types, Sorensen's similarity coefficient was calculated by the following formula

[40].

$$Ss = \frac{2a}{(2a + b + c)}$$

Where, Ss = Sorensen's similarity coefficient

 a=number of species common to both samples /communities/study areas

 b =number of species in sample1, c=number of species in sample 2

**2.5.3 Structural data analysis.** Density, basal area, frequency, IVI, and DBH class distributions were calculated for each woody plant species used for the description of vegetation structure. Thus, the following formulas were used to calculate the frequency, density, and basal area of woody species.

Species density refers to the number of individuals of plant species in a plot, measured in individuals per unit area. For tree density, this is calculated by converting the count from the total plots into a hectare basis [40]. The formula used to compute tree density is:

$$Density = \frac{\text{number of individuals of species A}}{\text{Sample area in hectare}}$$

$$Relative\ Density = \frac{\text{Density of species A}}{\text{Total density of all species}} \times 100$$

$$Frequency\ (F) = \frac{\text{number of plots in which species occur}}{\text{Total number of plots laid down in the study site}}$$

$$Relative\ frequency\ (RF)\ \frac{\text{Frequency of woody plant species}}{\text{Total frequency of woody plant specieste}} \times 100$$

$$Basal\ area\ (BA) = \frac{\text{DBH}}{4} \times 100, \text{where DBH is diameter at breast height}$$

$$Dominance = \frac{\text{Basal area of species A}}{\text{Area of plots in hectare}}$$

$$Relative\ Dominance = \frac{\text{Dominance for species A}}{\text{Total dominance of all species}} \times 100$$

An importance value index (IVI) was computed for all woody species based on their relative density (RD), relative dominance (RDO), and relative frequency (RF). This index is used to determine the overall importance of each species in the forest system [40].

The Importance Value Index (IVI) for each woody species was computed using the following formula.

$$IVI = RD + RDO + RF$$

**2.5.4 Regeneration status of the forest.** The regeneration status of the forest was analysed by comparing the density ratios between seedlings and mature individuals, seedlings and saplings, and saplings and mature individuals. When seedlings are greater than saplings and saplings greater than adults, the status has good regeneration, whereas seedlings are greater than

or less than saplings and saplings less than adults; the status is poor regeneration, and a species is present only in adult form it considered as not regenerating [41].

**2.5.6 Plant community classification.** Using the estimated value of percent cover for each woody species, Agglomerative hierarchical cluster analysis was performed in this work to classify the vegetation into plant community types using the free statistical program R version 4.12 [42]. Based on the relative magnitude of their synoptic values, one or two main indicator tree or shrub species were chosen for identifying the community. A total of 64 plant species and 52 sample plots were used in the analysis.

**2.5.7 Ordination.** A multivariate method called ordination is used to show interactions in a small-scale region between species, plots, and environmental variables [43]. Using SAS software, a preliminary analysis of the data was performed by computing Pearson's correlation coefficient to find significant correlations between environmental variables and remove auto-correlated variables from the ordination analysis. The present study used Canonical Correspondence Analysis (CCA) to look at the links between environmental variables and vegetation. Thirteen environmental factors (altitude, slope, aspect, soil moisture content, pH, sand, clay, silt, EC, total nitrogen, available phosphorus, CEC, and OM) as well as 52 sample plots and 64 species have been included in the analysis. To determine the effect of the environmental variables the distribution of plant communities, an Adonis test was performed. Cover abundance values for plant species and information on significant environmental variables were shown in the CCA ordination plot.

## 3. Results

### 3.1 Woody species composition

In the study area, 68 species of woody plants were identified, belonging to 44 families and 63 genera. Out of them, 64 species were found inside the plots, and 4 species—*Maytenus arbutifolia*, *Ficus sur*, *Lippia adoensis*, and *Combretum molle* were found outside the plot (S1 Appendix). Of the 64 species of woody plants that have been found, woody lianas make up the least represented growth form, accounting for 10.94% (7 in number) of the species. Of these, trees make up 48.44% (31 in number), and shrubs bring up 40.62% (26 in number) (S1 Appendix). In the study area, the plant families Asteraceae, Fabaceae, Euphorbiaceae and Rosaceae were the most diverse, each represented by 4 species, making up 6.25% of the total species observed (Table 1). The overall Shannon diversity index and evenness of the study area were 2.12 and 0.92, respectively (S2 Appendix).

### 3.2 Vegetation structure

The distribution of woody plant species in various DBH classes was analyzed and classified into five categories: DBH classes 1 = 2.5-10 cm, 2 = 10.1-20 cm, 3 = 20.1-30 cm, 4 = 30.1-40 cm, and 5 = > 40.1 cm. The first DBH class (2.5-10 cm) had a greater density of woody

**Table 1. Six families with the highest number of species.**

| Family | No. of Species | Percentage (%) |
|---|---|---|
| Asteraceae | 4 | 6.25% |
| Fabaceae | 4 | 6.25% |
| Euphorbiaceae | 4 | 6.25% |
| Rosaceae | 4 | 6.25% |
| Myrsinaceae | 3 | 4.69% |
| Celastraceae | 2 | 3.13% |

**Table 2. Distribution of woody plant species in different DBH classes Classes:1 = 2.5-10 cm, 2 = 10.1-20 cm, 3 = 20.1-30 cm, 4 = 30.1-40 cm, 5 = 40.1 - 50cm and 6 = > 50 cm in Seqela forest.**

| No | DBH class | No of individuals | density/ha |
|----|-----------|-------------------|------------|
| 1 | (2.5-10 cm) | 276 | 132.7 |
| 2 | (10.01-20 cm) | 235 | 113 |
| 3 | (20 -30 cm) | 177 | 85.1 |
| 4 | (30.1-40) | 119 | 57.2 |
| 5 | (> 40.1) | 72 | 34.6 |

individuals; 176 of them made up 28.8% of the total. 235 individuals, representing 24.5% of the total, were in the second DBH class (10.1–20 cm) density distribution. The density distributions for the remaining four classes were, in order, 177 (18.5%), 119 (12.4%), 72 (7.5%), and 80 (8.3%). From lower to higher DBH classes, the density distribution of woody individuals showed a declining trend (Table 2).

In the Seqela natural forest, the overall density of woody species per hectare was 1079.3. *Croton macrostachyus* (88.46 individuals per hectare) had the highest density, followed *by Calpurnia aurea* (68.27 individuals per hectare), *Albizia gummifera* (70.19 individuals per hectare), and *Olinia rochetiana* (78.37 individuals per hectare). *Albizia gummifera* (51.92%), *Croton macrostachyus* (44.23%), *Olinia rochetiana*, and *Teclea nobilis* (36.54%) were the most common woody species in the Seqela natural forest. *Erythrina brucei*, *Juniperus procera*, *Hagenia abyssinica*, *Myrsine africana*, *Laggera tomentosa*, and *Rumex nervosus* were the least common woody species, occurring in 1.92% of cases each. The total basal area of woody plant species with a DBH greater than 2.5 cm was 27.4 $m^2$ per hectare. The basal area of individual plant species increased as their DBH class increased, but their number decreased. Species such as *Erythrina brucei* (6.27 $m^2$/ha), *Prunus africana* (3.67 $m^2$/ha), *Vachellia abyssinica* (3.55 $m^2$/ha), and *Apodytes dimidiate* (2.59 $m^2$/ha) and *Schefflera abyssinica* (2.04 $m^2$/ha) contributed the largest basal area (Table 3).

### 3.3 Importance Value Index (IVI) of species

*Erythrina brucei* (IVI = 11.24), *Prunus africana* (8.68), Croton *macrostachyus* (IVI = 7.38), *Albizia gummifera* (IVI = 9.03), and *Vachellia abyssinica* (7.02) were the most dominant and ecologically significant species in the Seqela natural forest. Conversely, species like *Vernonia amygdalina*, *Juniperus procera*, *Hagenia abyssinica*, and *Dracaena steudneri* had lower IVI values (less than 1) (Table 4).

### 3.4 Regeneration status of woody plants in Seqela forest

The regeneration status of selected woody plant species in the Seqela forest was assessed to predict the forest's future fate. The regeneration status of the Seqela forest was classified as good

**Table 3. Basal areas of the top 6 dominant species in Seqela forest.**

| No | Scientific Name | Basa area ($m^2$/ha) |
|----|-----------------|----------------------|
| 1 | *Erythrina brucei* | 6.27 |
| 2 | *Prunus Africana* | 3.67 |
| 3 | *Vachellia abyssinica* | 3.55 |
| 4 | *Apodytes dimidiata* | 2.59 |
| 5 | *Albizia gummifera* | 2.05 |
| 6 | *Schefflera abyssinica* | 2.04 |

**Table 4. The highest and lowest (IVI) value of woody plant species in the study area.**

| NO. | Plant species Name | R.F | R.D | R.DO | IVI | % IVI |
|---|---|---|---|---|---|---|
| 1 | *Vachellia abyssinica* | 1.55 | 0.535 | 12.954 | 15.039 | 7.07* |
| 2 | *Albizia gummifera* | 5.232 | 6.503 | 7.49 | 19.225 | 9.03* |
| 3 | *Croton macrostachyus* | 4.457 | 8.196 | 3.047 | 15.7 | 7.38* |
| 4 | *Dracaena steudneri* | 0.388 | 0.222 | 0.42 | 1.03 | 0.48** |
| 5 | *Erythrina brucei* | 0.969 | 0.089 | 22.86 | 23.918 | 11.24* |
| 6 | *Hagenia abyssinica* | 0.193 | 0.089 | 0.0274 | 0.3094 | 0.15** |
| 7 | *Juniperus procera* | 0.193 | 0.044 | 1.104 | 1.341 | 0.63** |
| 8 | *Prunus africana* | 3.489 | 1.604 | 13.392 | 18.485 | 8.68* |
| 9 | *Vernonia amygdalina* | 0.775 | 0.357 | 0.128 | 1.26 | 0.59** |

* Indicates the highest value of IVI of woody plant species.

** Indicates the lowest value of IVI of woody plant species

regeneration, as seedlings (461.1 individuals/ha) > saplings (68.3 individuals/ha) > mature trees (44.7 individuals/ha) (Fig 3). Woody plant species such as *Schefflera abyssinica*, *Apodytes dimidiata*, *Erythrina brucei*, *Hagenia abyssinica*, *Juniperus procera*, *Myrica salicifolia*, and *Prunus africana* were not represented by both seedling and sapling stages. *Pittosporum viridiflorum*, *Vachellia abyssinica*, *Albizia gummifera*, *Buddleja polystachya*, *Clausena anisata*, *Croton macrostachyus*, *Dombeya torrida*, *Dovyalis abyssinica*, *Euphorbia abyssinica*, *Olea europaea* subsp. *cuspidata*, *Olinia rochetiana*, *Osyris quadripartita*, *Rhamnus prinoides*, *Rytigynia neglecta*, *Teclea nobilis*, and *Vernonia amygdalina* exhibited fair regeneration status. Conversely, woody species such as *Allophylus abyssinicus*, *Buddleja davidii*, *Discopodium penninervium*, *Dracaena steudneri*, *Ekebergia capensis*, *Grewia ferruginea*, *Maytenus obscura*, and *Rhus vulgaris* showed poor regeneration status.

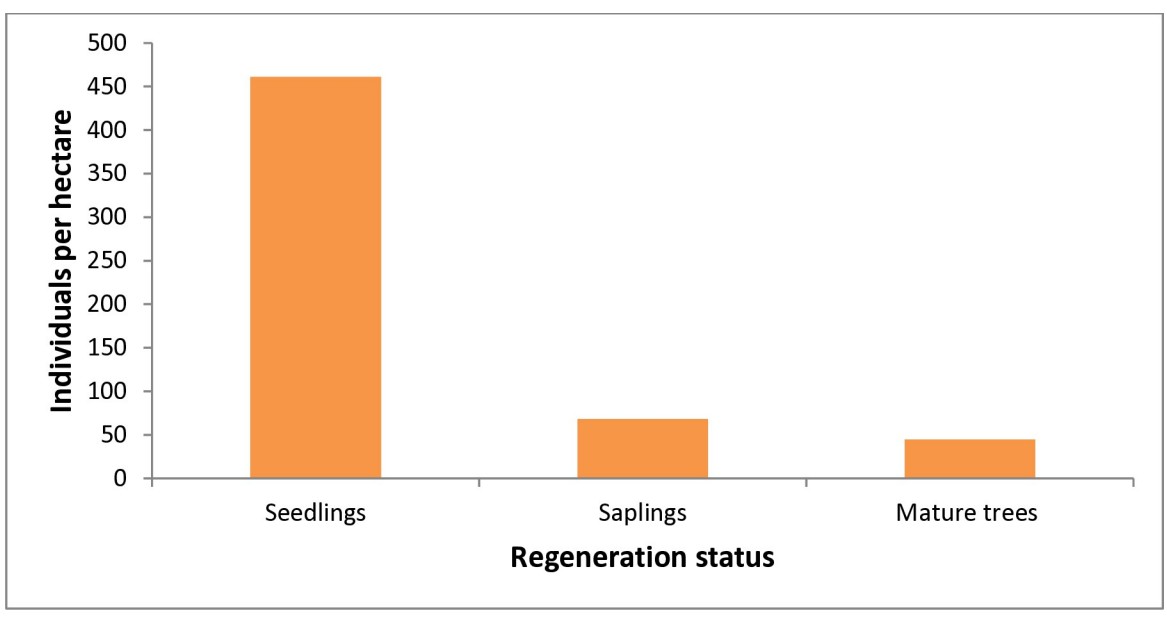

**Fig 3. Regeneration status of Seqela forest.**

### 3.5 Plant community types

Based on the agglomerative hierarchical cluster analysis performed in the study area, four plant community types were identified (Table 5). The cophenetic correlation coefficient reached a value of 0.774, indicating a considerable degree of similarity in the classification (Fig 4). These communities were grouped as follows: *Prunus africana - Croton macrostachyus; Albizia gummifer – Ekebergia capensis; Vachellia abyssinica - Dombeya torrida; Schefflera abyssinica - Teclea nobilis*.

### 3.6 *Albizia gummifera- Ekebergia capensis* Community Type I

This community type was distributed between altitudinal ranges of 2314 –2520 m.a.s.l. The dominant tree species are *Albizia gummifera*, *Ekebergia capensis*, *Croton macrostachyus*, *Erythrina brucei*, *Pittosporum viridiflorum* and *Olea europaea* subsp. *cuspidata*. The shrub species such as *Vernonia amygdalina*, *Maesa lanceolata*, *Maytenus addat*, *Clutia abyssinica*, *Rosa abyssinica*, *Vernonia myriantha* and *Bridelia micrantha* were highly dominant. The most dominant woody Lianas included in this community were *Zehneria scabra* and *Periploca linearifolia*. This community type is represented by 52 species, being the richest in a number of species among the four community types.

### 3.7 *Prunus africana- Croton macrostachyus* Community Type II

The altitudinal range of this community lies between of 2378 – 2441 m.a.s.l. The most dominant tree species were *Prunus africana*, *Olinia rochetiana*, *Apodytes dimidiata* and *Croton macrostachyus*. The shrub layer is dominated by *Osyris quadripartita*, *Calpurnia aurea*, *Asparagus africanus*, *Bersama abyssinica* and *Rytigynia neglecta*. Woody liana species such as *Jasminum abyssinicum*, *Urera hypselodendron*, *Clematis longicauda* and *Phytolacca dodecandra* were *dominant*. This community is comprised of 34 species.

### 3.8 *Vachellia abyssinica -Dombeya torrid* Community Type III

This community type was located between the altitudinal ranges of 2377 – 2577 m.a.s.l. The dominant plant species from this community were *Vachellia abyssinica*, *Dombeya torrida*, *Allophylus abyssinica*, *Maytenus obscura*, *Myrica salicifolia* and *Juniperus procera from* the tree layer and *Dovyalis abyssinica*, *Capparis tomentosa*, *Rubus steudneri*, *Rumex nervosus* and *Otostegia integrifolia* from shrub layer. This community is comprised of 48 species.

### 3.9 *Schefflera abyssinica - Teclea nobilis* Community Type IV

This community type was located between altitudinal ranges of 2335 – 2432 m.a.s.l. The dominant plant species in the community were *Schefflera abyssinica*, *Teclea nobilis*, *Euphorbia abyssinica*, *Dracaena steudneri*, *Solanecio gigas*, *Laggera tomentosa*, *Rhus glutinosa* and *Discopodium penninervium* from tree and shrub layer. This community is comprised of 27 species.

### 3.10 Species richness, evenness, and diversity of community types in the Seqela forest

We calculated the Shannon-Wiener diversity index for each of the four types of plant communities. As a result, according to Shannon's diversity, community 1 had the greatest species diversity and richness, followed by community 3, community 4 had the lowest. Conversely, communities 1 and 2 were followed in order of evenness value, with community 3 having the

**Table 5. Synoptic table of species.**

| Species Name | C-1 | C-2 | C-3 | C-4 |
|---|---|---|---|---|
| *Albiziagummifera* | **6** | 1.17 | 2.35 | 1 |
| *Ekebergiacapensis* | **5.12** | 1.17 | 0.88 | 0.42 |
| *Maesalanceolata* | 1.35 | 0 | 0 | 0.75 |
| *Carissaspinarum* | 1.47 | 0.33 | 0.71 | 0 |
| *Rosaabyssinica* | 1.12 | 0 | 0.94 | 0 |
| *Vernonia amygdalina* | 1.13 | 0 | 0 | 0 |
| *Vernoniamyriantha* | 1.12 | 1 | 0.18 | 0.17 |
| *Clutiaabyssinica* | 1.06 | 0.5 | 0.47 | 0 |
| *Erythrinabrucei* | 0.35 | 0 | 0 | 0 |
| *Maytenus addat* | 0.88 | 0.33 | 0 | 0 |
| *Brideliamicrantha* | 0.41 | 0.33 | 0.12 | 0 |
| *Hypericumquartinianum* | 0.35 | 0 | 0 | 0 |
| *Rhamnusprinoides* | 0.29 | 0 | 0.06 | 0 |
| *Zehneriascabra* | 0.18 | 0.17 | 0.06 | 0 |
| *Oleaeuropaea* | 0.71 | 0 | 0.12 | 0 |
| *Periploca linearifolia* | 0.29 | 0.17 | 0.12 | 0 |
| *Embeliaschimperi* | 0.18 | 0 | 0.12 | 0 |
| *Pittosporumviridiflorum* | 0.76 | 0 | 0.12 | 0 |
| *Bruceaantidysenterica* | 0.47 | 0.33 | 0 | 0.33 |
| *Gnidiaglauca* | 0.18 | 0 | 0.12 | 0 |
| *Crotonmacrostachyus* | 4.67 | **6.33** | 0.65 | 0 |
| *Prunusafricana* | 0.71 | **6.83** | 0.82 | 3.42 |
| *Oliniarochetiana* | 2 | 3.67 | 2.47 | 0 |
| *Calpurnia.aurea* | 1 | 2.33 | 0.88 | 0 |
| *Apodytesdimidiata* | 0.76 | 2.33 | 1.18 | 1.67 |
| *Osyrisquadripartita* | 0.53 | 1 | 0.47 | 0 |
| *Bersamaabyssinica* | 0.82 | 0.83 | 0 | 0.25 |
| *Galinierasaxifraga* | 0.47 | 0.67 | 0 | 0.17 |
| *Phytolaccadodecandra* | 0.18 | 0.33 | 0 | 0 |
| *Jasminumabyssinicum* | 0.29 | 0.5 | 0.12 | 0.08 |
| *Hageniaabyssinica* | 0 | 0.33 | 0 | 0 |
| *Clematis longicauda* | 0.29 | 0.33 | 0.18 | 0.25 |
| *Urera.hypselodendron* | 0.41 | 0.83 | 0.41 | 0.58 |
| *Asparagusafricanus* | 0 | 0.33 | 0.06 | 0 |
| *Rytigynianeglecta* | 0.24 | 1 | 0.24 | 0.67 |
| *Buddlejadavidii* | 0.06 | 0.83 | 0.71 | 0.42 |
| *Vachellia abyssinica* | 0 | 0 | **3.41** | 0 |
| *Dombeyatorrida* | 1.06 | 0.67 | **3.06** | 0.75 |
| *Allophylus abyssinicus* | 0.65 | 1.67 | 2.35 | 0 |
| *Clausenaanisata* | 1.24 | 0.83 | 1.29 | 0 |
| *Buddlejapolystachya* | 1 | 0 | 1.24 | 0.42 |
| *Dovyalisabyssinica* | 0.47 | 0.5 | 1.24 | 0 |
| *Myricasalicifolia* | 0.29 | 1 | 1.47 | 0 |
| *Maytenus obscura* | 0 | 0 | 1.82 | 0 |
| *Rhusvulgaris* | 0.71 | 0.17 | 0.82 | 0 |
| *Otostegiaintegrifolia* | 0.24 | 0 | 0.35 | 0 |
| *Acanthussennii* | 0.53 | 0 | 0.59 | 0 |

*(Continued)*

**Table 5.** (Continued)

| Species Name | C-1 | C-2 | C-3 | C-4 |
|---|---|---|---|---|
| *Capparistomentosa* | 0.12 | 0 | 0.29 | 0 |
| *Juniperusprocera* | 0 | 0 | 0.35 | 0 |
| *Rubussteudneri* | 0.35 | 0 | 0.53 | 0.17 |
| *Rumexnervosus* | 0 | 0 | 0.12 | 0 |
| *Grewiaferruginea* | 0.12 | 0 | 0.82 | 0 |
| *Dodonaeaangustifolia* | 0.12 | 0 | 0.29 | 0 |
| *Maytenusobscura* | 0 | 0 | 1.82 | 0 |
| *Myrsineafricana* | 0 | 0 | 0.12 | 0 |
| *Scheffleraabyssinica* | 0 | 0 | 0 | **6** |
| *Teclea nobilis* | 0.88 | 0.33 | 1.06 | **4** |
| *Discopodiumpenninervium* | 0.82 | 0 | 0.29 | 2.92 |
| *Solaneciogigas* | 0.35 | 1 | 0.12 | 1.5 |
| *Euphorbiaabyssinica* | 0.29 | 0 | 0 | 1.33 |
| *Ocimum lamiifolium* | 0 | 0 | 0.29 | 0.83 |
| *Laggeratomentosa* | 0 | 0 | 0 | 0.25 |
| *Rhusglutinosa* | 0 | 0 | 0 | 0.5 |
| *Dracaenasteudneri* | 0 | 0 | 0 | 0.17 |

greatest value. Among all types of communities, Community 4 had the lowest evenness scoring (Table 6).

### 3.11 Similarity among community types in the Seqela forest

To find the differences in species composition between the four community types, Sorensen's similarity coefficient (Ss) was calculated. Among all communities, the overall similarity coefficient varies from 31 to 45%. Communities 1 and 2 had the next-most similarity (43%) after C1 and C3 (45%). Relatively little similarity (31%) was found between Communities 3 and 4. (Table 7).

### 3.12 Relationship between plant community types and environment variables

The gradient length of 3.69 is greater than 3, as shown by the Detrended correspondence analysis (DCA) used to choose the optimal ordination technique. As a result, CCA was accepted as an adequate ordination for examining the interactions between the environmental variables and the vegetation. The longest DCA axis for the Sekela forest data set in the present study was 3.69, indicating the heterogeneity of the data as presented in (Table 8). To evaluate the connections between community type and environmental variables, a study using the unimodal response model is suggested. The Canonical correspondence analysis (CCA) ordination method is employed in this analysis.

### 3.13 CCA ordination

Thirteen environmental factors were taken into consideration in the ordination analysis: aspect, pH, phosphorus, sand, clay, silt, slope, nitrogen, moisture content, altitudeand Cation Exchange Capacity. As Table 9 and Fig 5 indicate, the only factors that significantly ($p < 0.05$) correlated with species compositions and their distributions were altitude, OM, slope, aspect, and phosphorus. The canonical correlation analysis (CCA) diagram showed that aspect, av P,

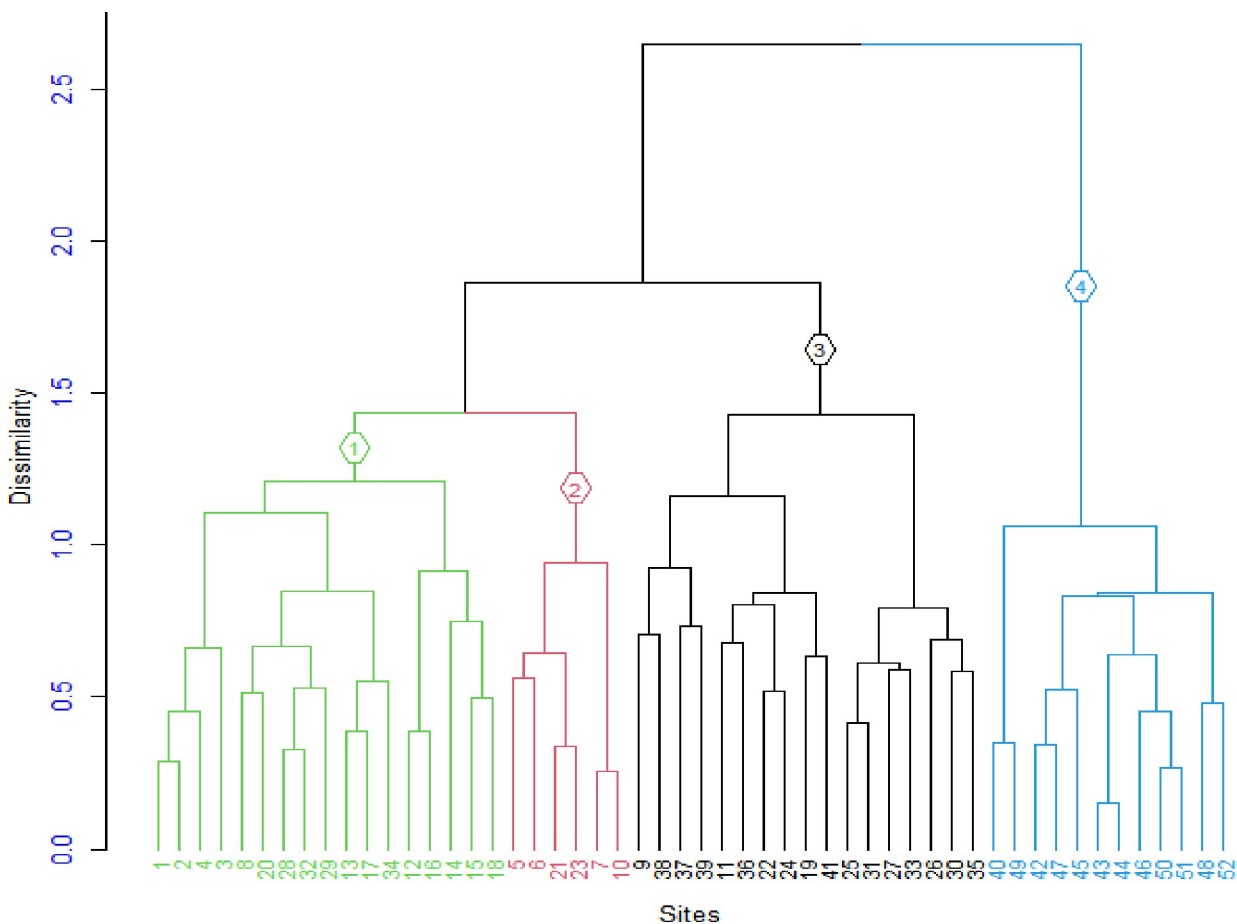

**Fig 4. Dendrogram of the vegetation data obtained from agglomerative hierarchical cluster analysis of the study area (Community type 1, 2, 3, and 4).**

and OM were the main variables related to the first axis, and altitude and slope were the major factors associated with the second axis (Fig 5 and Table 10). The first and second axes' respective eigenvalues were 0.34 and 0.22. For the first and second axes, there was heterogeneity in patterns of plant species distribution and plant community formation of 41% and 22.9%, respectively. The results showed that the first two axes accounted for 63.9% of the variation in patterns of plant species distribution and plant community formation. Thus, it was sufficient to reflect the relationship between species and environmental factors using the first two axes.

## 4. Discussion

### 4.1 Woody species composition

In the Seqela forest, we found 68 woody species. This lower than other studies conducted in other dry Afromontane forests of Ethiopia, such as those conducted in the Zege Peninsula

**Table 6. Species Richness, diversity and evenness of the communities.**

| Community type | Richness | Diversity (H) | Shannon evenness |
|---|---|---|---|
| 1 | 52 | 3.39 | 0.863 |
| 2 | 34 | 2.99 | 0.847 |
| 3 | 48 | 3.35 | 0.865 |
| 4 | 27 | 2.64 | 0.801 |

**Table 7. Sorensen's similarity of the communities.**

**Community types**

| -   | C-1  | C-2  | C-3  | C-4 |
|-----|------|------|------|-----|
| C-1 | -    |      |      |     |
| C-2 | 0.43 | -    |      |     |
| C-3 | 0.45 | 0.41 | -    |     |
| C-4 | 0.38 | 0.34 | 0.31 | -   |

Note: C-1: Albizia gummifer-Ekebergia.capensis, C-2: Prunus Africana-Croton macrostachyus, C-3: Vachellia abyssinica -Dombeya torrida and C-4: Schefflera abyssinica -vepris nobilis community types.

forest [44] with 113 species, Tara Gedam and Abebaye forests [45] with 143 species, and the forest patches in the Dega Damot district [22] with 106 woody species, the study area showed lower species richness in woody species. The study area's species richness was higher than that of other dry Afromontane forests in Ethiopia, like the 64-species Denkoro Forest in Wello [46], the 50-species Zengena Forest [44], the 66-species Kuandisha Forest [47], and the 57-species Amoro Forest [48].

Asteraceae, Fabaceae, Euphorbiaceae, and Rosaceae were among the plant families that were the most common in the study area. The order of families in terms of species richness is consistent with previous studies on dry evergreen Afromontane forests, which indicated the dominance of Asteraceae [49, 50]. Similarly, [47] reported that the most frequent family in the Kuandisha dry evergreen Afromontane forest fragment was Fabaceae. According to [51], the two most prominent plant families in the flora of Ethiopia and Eritrea are the Asteraceae and Fabaceae. In Seqela Forest, the dominance of Asteraceae and Fabaceae may suggest that disturbances have occurred in the forest. Asteraceae plants are typically ruderal and thrive best in open, disturbed soil [30]. According to [30], the ability of Fabaceae to fix atmospheric nitrogen helps them to survive on soils deficient in nutrients, which accounts for their success in taking over disturbed habitats.

The overall Shannon diversity index for the study area forest patches was 2.12, which is lower than other dry evergreen Afromontane forests such as the Tara Gedam forest (H' = 2.98)) [45]. and Zege forest (H' = 3.72) [44]. On the other hand, it is higher than in another forest eg. Abebaye forest (H' = 1.31) [45]. This variation might be due to the altitudinal range, size of the forests, degree of disturbance, and habitat diversity. [41] reported that the Shannon-Wiener diversity index normally varies between 1.5 and 3.5 and rarely exceeds 4.5. Thus, the overall diversity of the forest of the study area falls within the normal range of Shannon-Wiener diversity.

## 4.2 Vegetation structure

The woody plant species distribution in Seqela Forest showed an inverted J-shape pattern, suggesting that small to medium-sized DBH trees dominated the forest rather than large-sized trees. This could be because local communities used these species for various purposes.

**Table 8. Detrended correspondence analysis results of Seqela forest.**

|                 | CC1    | CC2    | CC3    | CC4    |
|-----------------|--------|--------|--------|--------|
| Eigenvalues     | 0.5231 | 0.3083 | 0.2267 | 0.2275 |
| Decorana values | 0.5451 | 0.3056 | 0.2090 | 0.1766 |
| Axis lengths    | 3.6875 | 2.5179 | 2.9114 | 1.9855 |

**Table 9. Results of function donis test of environmental variables (significant environmental variables are indicated by Asterix at their p-value).**

| Environmental variables | Df | Sums of Sqs. | Mean Sqs | F. Model | $R^2$ | Pr (>F) |
|---|---|---|---|---|---|---|
| Slope (%) | 1 | 0.7574 | 0.75742 | 3.4429 | 0.04864 | 0.01 ** |
| Aspect | 1 | 2.8341 | 2.83412 | 12.8827 | 0.18199 | 0.01 ** |
| Altitude (m) | 1 | 0.5114 | 0.51141 | 2.3246 | 0.03284 | 0.03* |
| MC | 1 | 0.2986 | 0.29864 | 1.3575 | 0.01918 | 0.19 |
| Sand | 1 | 0.3544 | 0.35440 | 1.6109 | 0.02276 | 0.09. |
| Silt | 1 | 0.1304 | 0.13041 | 0.5928 | 0.00837 | 0.74 |
| Clay | 1 | 0.2647 | 0.26471 | 1.2033 | 0.01700 | 0.18 |
| PH | 1 | 0.2393 | 0.23929 | 1.0877 | 0.01537 | 0.33 |
| EC | 1 | 0.1350 | 0.13501 | 0.6137 | 0.00867 | 0.75 |
| OM | 1 | 0.5804 | 0.58039 | 2.6382 | 0.03727 | 0.02 * |
| CEC | 1 | 0.1748 | 0.17482 | 0.7947 | 0.01123 | 0.67 |
| N | 1 | 0.3728 | 0.37280 | 1.6946 | 0.02394 | 0.10 |
| P | 1 | 0.5595 | 0.55946 | 2.5431 | 0.03593 | 0.02 * |
| Residuals | 38 | 8.3598 | 0.21999 | 0.53683 | | |
| Total | 52 | 15.5727 | | | | 1.000 |

Signif.codes: 0 '***' 0.001

'**' 0.01

'*' 0.05 '.' 0.1 ' '

The total density of woody species in the Seqela forest was 1079.3 individuals per hectare. This was less than the results of studies conducted by [44] in the Zegie Peninsula Forest (3318 individuals per hectare), [47] in the Kundasha Forest (3086 individuals per hectare), and [48] in the Amoro Forest (2860 individuals per hectare). The density was also higher than other forests, such as Wanzaye natural forest (498) [49]. The degree of anthropogenic interference in various forests may account for this variation in density. In the research area, species with the highest density values were *Clutia abyssinica*, *Croton macrostachyu*s, *Albizia gummifera*, *Olinia rochetiana*, and *Calpurnia aurea*. This might be because the species is small, resulting to a greater total count within a given plot size and a fairly excellent ability for regeneration in dry situations under shade. On the other hand, low-density species may be associated with increased grazing and tree and shrub cutting for charcoal and firewood [50]. The most common species in Seqela woodland were *Albizia gummifera*, *Croton macrostachyus*, *Olinia rochetiana*, *Urera hypselodendron*, and *Teclea nobilis*. These species were pioneer species, as reported by [45] *Albizia gummifera* and *C. macrostachyus* were also the most frequent species in Tara Gedam and Abebaye forests, respectively. The total basal area of the Seqela forest was 27.4 m$^2$ per hectare, and the basal area of individual plant species increased as their DBH class increased but their number decreased. The study area had relatively less total basal area than some dry Afromontane forests in Ethiopia, such as Tara Gedam Forest and Abebaye forests [45] and Wof washa forest [28], and Dega Damot district forest patches [39]. On the other hand, the basal area was greater than other forests, such as Amoro forest [48] and Kundasha forest [47]. The total basal area of the study area was comparable to that of the Zengena forest [44]. Species such as *Erythrina brucei*, *Prunus africana*, *Vachellia abyssinica*, *Apodytes dimidiate*, *Albizia gummifera*, and *Schefflera abyssinica* had high basal area due to their large size, even though they had low density. The Importance Value Index (IVI) value was an important parameter used to determine the significance of species in a given ecosystem for conservation purposes [54]. Species with lower IVI values required high conservation efforts [55], while those with higher IVI values needed monitoring and management [53].

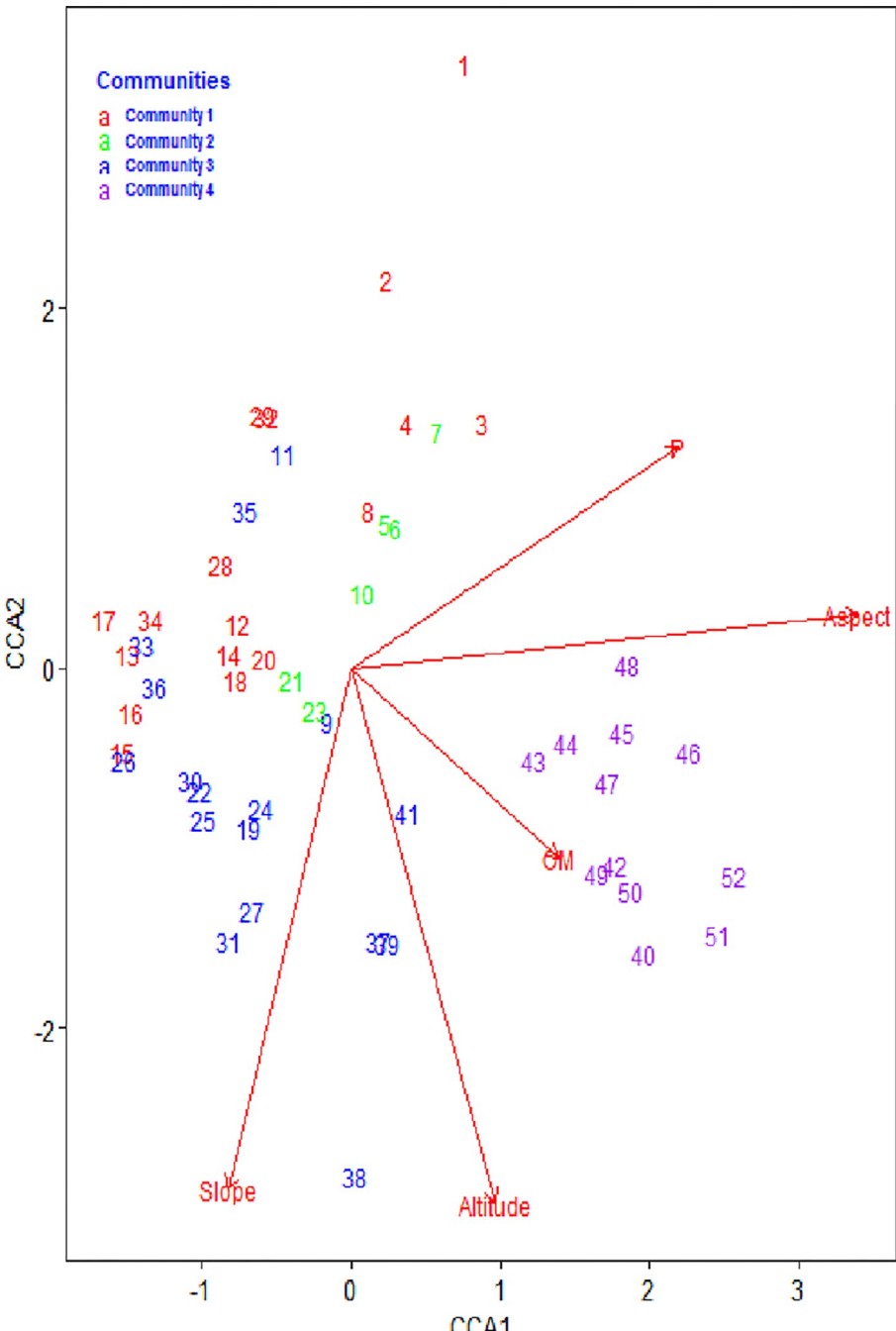

**Fig 5. CCA ordination graph, significant environmental variables (p< 0.05), and plant community in the study area.**

## 4.3 Regeneration status

The regeneration status of natural vegetation is considered good regeneration if seedlings are greater than saplings and saplings are greater than adults [41]. In the case of the Seqela forest, the regeneration status was considered as good regeneration since seedlings (461.1 individuals per hectare) > saplings (68.3 individuals per hectare) >mature trees (44.7 individuals per

**Table 10. Bi-plot scores for constrained variables.**

| Variables | CCA1 | CCA2 | CCA3 | CCA4 |
|---|---|---|---|---|
| Slope | -0.22 | -0.785 | -0.251 | 0.44660 |
| Aspect | 0.92 | 0.081 | 0.241 | -0.27716 |
| Altitude | 0.26 | -0.807 | 0.480 | 0.16719 |
| OM | 0.38 | -0.288 | 0.073 | 0.00035 |
| P | 0.60 | 0.334 | 0.162 | 0.62852 |
| Eigenvalue | 0 .3864 | 0.2149 | 0.1497 | 0.1043 |
| Proportion Explained | 0.4121 | 0.2292 | 0.1597 | 0.1112 |
| Cumulative Proportion | 0.4121 | 0.6413 | 0.8010 | 0.9122 |

hectare). The possible reasons for the existence of good regeneration status in the forest might be a diverse and abundant seed bank in the soil can ensure a continuous supply of seeds for regeneration.

## 4.4 Plant communities

In the Seqela forest, four plant communities were identified based on the findings of the hierarchical cluster analysis. However, species diversity and richness differed among plant communities. The differences in species composition along the communities may be due to the variation in physico-chemical properties of soil, topography, climate, weathering processes, and microbial activities [56–58] and several other biotic and abiotic factors [14]. For instance, communities 1 and 3 exhibited the highest species richness and diversity. Conversely, community 4 had the lowest species richness and diversity compared to the other communities. The reason for the high species diversity and richness in communities 1 and 3 may be due to the highest altitudinal ranges in which these communities are found, ranging from 2378 to 2441 and 2377 to 2577 m.a.s.l., respectively [39]. Additionally, community 4 was the most disturbed community, likely due to agricultural expansion and overgrazing by livestock.

## 4.5 Relationship between plant community types and environment variables

The topography (aspect, slope, and altitude) and soil variables combined together to shape the distribution of plant communities in the Seqela forest. The main environmental factors that explained variations in plant species distribution and patterns of plant community formation in the research area were aspect, slope, altitude, OM, and av P, according to the CCA results. According to [56], aspect was shown to be the most significant environmental variable in determining vegetation variation. This could be attributed to variations in incoming solar radiation that impact temperature and moisture levels. Plant growth and the creation of plant communities are strongly influenced by the temperature and moisture of the soil. Slope and altitude were the primary factors influencing vegetation distribution, according to similar studies conducted in other dry Afromontane forests in Ethiopia [48, 59]. A similar study by [22] also confirms that organic matter and av-P were the most important constraining variables in plant community formation of the study area.

## 5. Conclusions

The woody species composition of the Seqela forest indicates a good level of woody species richness in the area. Four community types, including *Albizia gummifera - Ekebergia capensis*, *Prunus Africana* and *Croton macrostachyus*, *Vachellia abyssinica* and *Dombeya torrida*, -

*Schefflera abyssinica - Teclea nobilis*, were identified. Canonical correspondence analysis (CCA) showed that aspect, altitude, slope, OM, and P were the primary factors of plant distribution patterns and plant community types. This suggests that the distribution of plant species and the formation of plant communities are heavily reliant on these environmental conditions. Species with high densities at lower DBH classes were characteristic of the general vegetation structure of Seqela forest; this may be because shrubby species dominated. The assessment of the forest's regeneration status indicated a good degree of regeneration, as evidenced by the presence of both seedlings and saplings for several species. However, some species, such as *Schefflera abyssinica*, *Apodytes dimidiata*, *Erythrina brucei*, *Hagenia abyssinica*, *Juniperus procera*, *Myrica salicifolia*, and *Prunus africana*, were observed without seedlings or saplings. These variations in regeneration among species reflect the natural dynamics of the forest.

## Supporting information

**S1 Appendix. List of woody plant species collected from Seqela forest Quarit district.** (DOCX)

**S2 Appendix. The values for diversity (H) and evenness are as follows for each plots.** (DOCX)

## Acknowledgments

The authors wish to acknowledge the Agriculture and Rural Development Office of the Quarit District and the respective chairpersons for their support. Special thanks are extended to the University of Leicester's Institute for Environmental Futures and Space Park Leicester for providing office space, internet access, and reference materials that facilitated this research. The authors also acknowledge H.B.'s financial support from the Natural Environment Research Council (NERC) via the National Centre for Earth Observation (NCEO), which contributed to these logistical provisions.

## Author Contributions

**Conceptualization:** Getaneh Moges.

**Data curation:** Liyew Birhanu, Getaneh Moges.

**Formal analysis:** Liyew Birhanu.

**Methodology:** Liyew Birhanu.

**Resources:** Getaneh Moges, Nigussie Amsalu, Heiko Balzter.

**Software:** Liyew Birhanu.

**Supervision:** Liyew Birhanu.

**Writing – original draft:** Liyew Birhanu, Getaneh Moges.

**Writing – review & editing:** Liyew Birhanu, Nigussie Amsalu, Heiko Balzter.

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
