## [Decision Letter · Decision Letter 0]

27 Feb 2024

PONE-D-24-05897Plant community distribution along environmental gradient of Seqela Dry Afromontane Forest in Northwestern EthiopiaPLOS ONE

Dear Dr. Birhanu,

Thank you for submitting your manuscript to PLOS ONE. After careful consideration, we feel that it has merit but does not fully meet PLOS ONE’s publication criteria as it currently stands. Therefore, we invite you to submit a revised version of the manuscript that addresses the points raised during the review process.

**ACADEMIC EDITOR: **The paper needs a major revision.==============================

We look forward to receiving your revised manuscript.

Kind regards,

Faham Khamesipour, Ph.D.

Academic Editor

PLOS ONE

Journal Requirements:

3. We noticed you have some minor occurrence of overlapping text with the following previous publication(s) among others, which needs to be addressed:

https://ecologicalprocesses.springeropen.com/articles/10.1186/s13717-020-00257-2?

In your revision ensure you cite all your sources (including your own works), and quote or rephrase any duplicated text outside the methods section. Further consideration is dependent on these concerns being addressed.

We require you to either (1) present written permission from the copyright holder to publish these figures specifically under the CC BY 4.0 license, or (2) remove the figures from your submission

Additional Editor Comments (if provided):

The paper needs a major revision. English needs improvement. The reviewers made more corrections and suggestions directly in the manuscript.

Reviewers' comments:

Reviewer's Responses to Questions

**Comments to the Author**

1. Is the manuscript technically sound, and do the data support the conclusions?

Reviewer #1: Partly

Reviewer #2: Yes

2. Has the statistical analysis been performed appropriately and rigorously? 

Reviewer #1: Yes

Reviewer #2: Yes

3. Have the authors made all data underlying the findings in their manuscript fully available?

Reviewer #1: Yes

Reviewer #2: Yes

4. Is the manuscript presented in an intelligible fashion and written in standard English?

Reviewer #1: Yes

Reviewer #2: Yes

5. Review Comments to the Author

Reviewer #1: Manuscript Number: PONE-D-24-05897

Plant community distribution along environmental gradient of Seqela Dry Afromontane

Forest in Northwestern Ethiopia

PLOS ONE

General comments

The research work presents interesting results but the manuscript needs to improve the comments provided below. Botanical methods of plant species identification and deposition herbarium with voucher numbers are not clearly stated and need to be addressed before any further consideration. Furthermore, the title needs to be rethought. In the current state, it is not very informative.

Specific comments:

Title: I would suggest that your title needs to be modified in a way that it should accurately describe the contents of your manuscript. Because the phrase “Plant community distribution along environmental gradient” is not inclusive but rather a single part of your paper.

Abstract

Line 2: “human population growth” is an equivocal reason even if it will be an umbrella driver for the anthropogenic factors mentioned formerly.

Line 7: “100 m”, is it altitudinal or geographical distance?

Line 12: “A total of 68 woody plant species”, why not include herbaceous species according to your title?

Line 14: Replace “Prunus Africana” with “Prunus africana” see in Table 3. Additionally, check the revised taxonomic name of “Acacia abyssinica”?

Keywords/phrases: I would suggest to the authors that it is better to focus on specific and relevant words to your content rather than broad and generic long phrases like “Canonical Correspondence Analysis, Hierarchical cluster analysis”.

Introduction

In paragraph 1, the first two statements: The authors wrote the role of plant resources in the Ethiopian context but cited unrelated authors (Fraticelli et al., 2013)?

Lack of connectivity and flow of ideas between statements such as hypotheses and objectives.

Methodology

Study area

See the legend in your map and the caption here: “Figure 1. Sekela forest is located in the Quarit district of West Gojjam Zone, Amhara Region in Ethiopia” Which is correct “Sekela” or Seqela”?

“Sekela Forest is categorised as dry evergreen Afromontane Vegetation”, this needs to be rephrased because it contradicts with demonstration of the cliamadiagram in Figure 2.

Be consistent in your reference citation… See “Friis et al. (2010)” … “(Asaminew Tassew et al., 2011)”.

What is your reason for your selection of the sub quadrats 5 × 5 m2 for the collection of seedling and sapling plants?

Vegetation data collection

I find the sampling design rather vague.

Botanical methods, typically herbarium procedure are missed out.

Data Analysis

Plant community classification

There is a mix of information herein: “agglomerative hierarchical cluster analysis”? In your result section: “Based on the hierarchical cluster analysis performed in the study area, four plant community types were identified (Figure 3).”

Where is the anthropogenic factor analysis, which has been mentioned as a major threatening factor?

Diversity and similarity indices

Diversity is okay but where is the similarity index?

Results

Line 3: Replace “Cobretum molle” with “Combretum molle”

Plant community types

Replace “Ekebergia.capensis” with “Ekebergia capensis”

“Figure 3. Dendrogram”, here is my big question why not 2 types or three community types or else? Because the author did not show the value of the cophenetic correlation coefficient where they determined the cutree.

Community type I.

Replace “Olea europaea subsp. Cuspidate” with “Olea europaea subsp. cuspidata”

Community type II

“Phytolacca dodecandra were dominant dodecandra” needs to be revised.

Similarity among community types in the Sekela Forest

“To find the differences in species composition between the four community types, Sorensen's

similarity coefficient (Ss) was calculated.” My concern here: I did not see this part in the method section.

Figures 3, 4 and 5 need to increase their resolutions.

References

Please correct the font type “Alelign A , Teketay D, Yemshawy, & Sue Edwards (2007)”.

Reviewer #2: The following suggestions may help to improve the quality of the Ms. further. Overall, the title clearly reflects the contents. The introduction does establish the existing state of knowledge but needs minor revision as suggested.

Authors should add the knowledge gap at the end of the introduction with proper citations.

The set of measurements was comprehensive. Discussion part needs revision as suggested in the text.

The conclusion needs concreteness. Do not repeat the results in the conclusions.

Figure 5 needs proper drawing (reduce the size appropriately)

The citations need up-dated.

English needs improvement.

I have made more corrections and suggestions directly in the manuscript.

Please revise the paper accordingly.

6. PLOS authors have the option to publish the peer review history of their article (what does this mean?). If published, this will include your full peer review and any attached files.

Reviewer #1: No

Reviewer #2: No

---

## [Author Response · Author response to Decision Letter 0]

1 Apr 2024

Manuscript number: PONE-D-24-05897

Title: Plant community distribution along environmental gradient of Seqela Dry Afromontane Forest in Northwestern Ethiopia

Journal Name: PLOS ONE

Article type: Research Paper

We would like to thank the reviewers for their efforts in reviewing the manuscript and providing valuable suggestions for improving the article. All comments and suggestions were addressed. The changes are marked in yellow color as shown in the revised manuscript. We detailed response for each comment is provided below. 

Editor 

Dear Dr. Birhanu,

Thank you for submitting your manuscript to PLOS ONE. After careful consideration, we feel that it has merit but does not fully meet PLOS ONE’s publication criteria as it currently stands. Therefore, we invite you to submit a revised version of the manuscript that addresses the points raised during the review process.

ACADEMIC EDITOR:

The paper needs a major revision.

Author’s Response: Thanks a lot for letting us revise our paper. Below are the responses to all reviewer comments.

Reviewer #1:

General comments 

1. The research work presents interesting results but the manuscript needs to improve the comments provided below. Botanical methods of plant species identification and deposition herbarium with voucher numbers are not clearly stated and need to be addressed before any further consideration. Furthermore, the title needs to be rethought. In the current state, it is not very informative.

Author’s Response: Thanks a lot for this positive feedback. We are sorry that some of the information about Botanical methods of plant species identification and deposition herbarium was not included, and we have improved this in the revised manuscript.

Specific comments:

2. Title: I would suggest that your title needs to be modified in a way that it should accurately describe the contents of your manuscript. Because the phrase “Plant community distribution along environmental gradient” is not inclusive but rather a single part of your paper.

3. Author’s Response: We appreciate the reviewer observation. The title was corrected as Woody species diversity, Vegetation structure and Plant community distribution along environmental gradients of Seqela Dry Afromontane Forest in Northwestern Ethiopia

Abstract

4. Line 2: “human population growth” is an equivocal reason even if it will be an umbrella driver for the anthropogenic factors mentioned formerly. 

Author’s Response: Thank you for this suggestion.

5. Line 7: “100 m”, is it altitudinal or geographical distance?

6. Line 14: Replace “Prunus Africana” with “Prunus africana” see in Table 3. Additionally, check the revised taxonomic name of “Acacia abyssinica”?

 Author’s Response: Again, we appreciate the reviewer observation reviewer. We have corrected the genus Acacia to Vachellia in the revised version of the manuscript.We have corrected as Vachellia abyssinica

7. /phrases: I would suggest to the authors that it is better to focus on specific and relevant words to your content rather than broad and generic long phrases like “Canonical Correspondence Analysis, Hierarchical cluster analysis”.

Author’s Response: Thanks for this suggestion and improved clarity of the paper in the revised document. 

Introduction

8. In paragraph 1, the first two statements: The authors wrote the role of plant resources in the Ethiopian context but cited unrelated authors (Fraticelli et al., 2013)

 Author’s Response: Thank you for this suggestion.

Methodology

Study area

9. See the legend in your map and the caption here: “Figure 1. Sekela forest is located in the Quarit district of West Gojjam Zone, Amhara Region in Ethiopia” Which is correct “Sekela” or Seqela”?

 Author’s Response: Seqela is the correct one.The correction has been made in the revised manuscript

10. Seqela Forest is categorised as dry evergreen Afromontane Vegetation”, this needs to be rephrased because it contradicts with demonstration of the cliamadiagram in Figure 2. Author’s Response:Olea europaea subsp. cuspidata, Juniperus procera, and Podocarpus falcatus are indicator species to the dry Afromontane forest. Therefore, Olea europaea subsp. Cuspidate species is found in the Seqela forest. Actually the vegetation type classification based on elevation, climate and indicator species…. Thus it is categorised as dry evergreen Afromontane forest and grassland complex (DAF).

11. Be consistent in your reference citation… See “Friis et al. (2010)” … “(Asaminew Tassew et al., 2011)”.

Author’s Response: Thank you very much for your excellent observation. We have corrected any inconsistency written of reference citation

Vegetation data collection

12. I find the sampling design rather vague. Botanical methods, typically herbarium procedure are missed out.

Author’s Response: Thanks for this suggestion and improved clarity of the paper in the revised manuscript. 

Data Analysis

Plant community classification 

13. There is a mix of information herein: “agglomerative hierarchical cluster analysis”? In your result section: “Based on the hierarchical cluster analysis performed in the study area, four plant community types were identified (Figure 3).”

Author’s Response: Agglomerative hierarchical cluster analysis. We have corrected the inconsistency writing of this word. 

14. Where is the anthropogenic factor analysis, which has been mentioned as a major threatening factor?

Author’s Response: No anthropogenic data collection and analysis in our study.

Diversity and similarity indices

18. Diversity is okay but where is the similarity index?

Author’s Response: We have added the information in the method section of revised manuscript. 

Results 

19. Line 3: Replace “Cobretum molle” with “Combretum molle”

Author’s Response: Corrected in the revised manuscript.

Plant community types

20. Replace “Ekebergia.capensis” with “Ekebergia capensis”

Author’s Response: Corrected in the revised manuscript.

21. “Figure 3. Dendrogram”, here is my big question why not 2 types or three community types or else? Because the author did not show the value of the cophenetic correlation coefficient where they determined the cutree. 

Author’s Response: We appreciate the reviewer observation. We agree that before doing cluster analysis, we need to determine the optimum cluster number (K). Among the methods we use to decide the optimum cluster are Elbow, silhouette, gap statistics and K-means partitioning using cascadeKM (calinski) and Similarity Ratio. We have access to scripts for each method. We tried to check the matrified vegetation data with elbow method and simple visual observation during the field work. Accordingly, we have done full analysis and found 4 plant community types. BTW, methods use to decide the optimum cluster alone may not show a clear classification. So visual observation during the field work also helps.

Community type I.

Replace “Olea europaea subsp. Cuspidate” with “Olea europaea subsp. cuspidata”

Author’s Response: corrected in the revised manuscript.

Community type II

“Phytolacca dodecandra were dominant dodecandra” needs to be revised.

Author’s Response: corrected

Similarity among community types in the Seqela Forest

“To find the differences in species composition between the four community types, Sorensen's similarity coefficient (Ss) was calculated.” My concern here: I did not see this part in the method section.

Author’s Response: We have added the information in the method section of revised manuscript 

Figures 3, 4 and 5 need to increase their resolutions.

Author’s Response: We have tried to clearly drown and describe the caption of the figure under the diagram and check the revised manuscript.

References

Please correct the font type “Alelign A , Teketay D, Yemshawy, & Sue Edwards (2007)”.

Author’s Response: The correction has been made in the revised manuscript

Reviewer #2: The following suggestions may help to improve the quality of the Ms. further. Overall, the title clearly reflects the contents. The introduction does establish the existing state of knowledge but needs minor revision as suggested. Authors should add the knowledge gap at the end of the introduction with proper citations. The set of measurements was comprehensive. Discussion part needs revision as suggested in the text.

The conclusion needs concreteness. Do not repeat the results in the conclusions. Figure 5 needs proper drawing (reduce the size appropriately).The citations need up-dated.

English needs improvement. I have made more corrections and suggestions directly in the manuscript. Please revise the paper accordingly. Please note that Introduction should be started with the general statement not by specific first. As it is an International paper and will be viewed by international audience not only by Ethiopian’s.

Author’s Response: Thank you for your efforts in reviewing the manuscript and providing valuable suggestions for improving the article. You have corrected many things of our manuscript. We have tried to clearly correct the manuscript and check the revised manuscript.

---

## [Decision Letter · Decision Letter 1]

4 Apr 2024

PONE-D-24-05897R1Woody species diversity, Vegetation structure and Plant community distribution along environmental gradients of Seqela Dry Afromontane Forest in Northwestern EthiopiaPLOS ONE

Dear Dr. Birhanu,

Thank you for submitting your manuscript to PLOS ONE. After careful consideration, we feel that it has merit but does not fully meet PLOS ONE’s publication criteria as it currently stands. Therefore, we invite you to submit a revised version of the manuscript that addresses the points raised during the review process. **The authors have appropriately revised the manuscript and answered all the questions of the reviewers satisfactorily.**

**I recommend the revised version of the manuscript be accepted for publication in PLOS One after minor revision.**

We look forward to receiving your revised manuscript.

Kind regards,

Faham Khamesipour, Ph.D.

Academic Editor

PLOS ONE

Journal Requirements:

Additional Editor Comments:

The authors have appropriately revised the manuscript and answered all the questions of the reviewers satisfactorily.

I recommend the revised version of the manuscript be accepted for publication in PLOS One after minor revision.

Reviewers' comments:

Reviewer's Responses to Questions

**Comments to the Author**

1. If the authors have adequately addressed your comments raised in a previous round of review and you feel that this manuscript is now acceptable for publication, you may indicate that here to bypass the “Comments to the Author” section, enter your conflict of interest statement in the “Confidential to Editor” section, and submit your "Accept" recommendation.

Reviewer #1: (No Response)

Reviewer #2: All comments have been addressed

2. Is the manuscript technically sound, and do the data support the conclusions?

Reviewer #1: Yes

Reviewer #2: Yes

3. Has the statistical analysis been performed appropriately and rigorously? 

Reviewer #1: Yes

Reviewer #2: Yes

4. Have the authors made all data underlying the findings in their manuscript fully available?

Reviewer #1: Yes

Reviewer #2: Yes

5. Is the manuscript presented in an intelligible fashion and written in standard English?

Reviewer #1: Yes

Reviewer #2: Yes

6. Review Comments to the Author

**Reviewer #1:** Manuscript Number: PONE-D-24-05897R1

Woody species diversity, Vegetation structure and Plant community distribution along environmental gradients of Seqela Dry Afromontane Forest in Northwestern Ethiopia

PLOS ONE

General comments

The authors have satisfactorily addressed the majority of my concerns and comments, although some minor ones still need to be addressed.

Title: It needs minor revision “Woody species diversity, structure, and community distribution along environmental gradients of Seqela Dry Afromontane Forest in Northwestern Ethiopia”

Introduction

In paragraph 2, replace “[25; 26]” with “[25, 26].

“… environmental factors [e.g. 27- 31]” please restate this statement. Additionally, use proper punctuation and writing techniques throughout your paper such as, replace “27- 31” with “27–31”.

Methodology

Study area

Paragraph 2 replace “Eragrostis teff” with “Eragrostis tef”.

Data Analysis

Plant community classification

##There is a mix of information herein: “agglomerative hierarchical cluster analysis”? Select either “agglomerative cluster analysis” or “hierarchical cluster analysis” with acceptable reason.

##The author did not give sufficient response for this question raised in the first draft.

Results

Plant community types

## “Figure 3. Dendrogram”, here is my big question why not 2 types or 3 community types or else? Because the author did not show the value of the cophenetic correlation coefficient where they determined the cutree.

##The author did not give sufficient response for this question raised in the first round of review rather they replied very subjective suggestions.

**Reviewer #2:** Dear Author(s)

Thank you for submitting the revised manuscript. You have appropriately revised the manuscript and answered all the questions of the reviewers satisfactorily.

I recommend the revised version be accepted for publication in PLOS One.

7. PLOS authors have the option to publish the peer review history of their article (what does this mean?). If published, this will include your full peer review and any attached files.

Reviewer #1: No

Reviewer #2: No

---

## [Author Response · Author response to Decision Letter 1]

6 Apr 2024

Manuscript Number: PONE-D-24-05897R2

Woody species diversity, Vegetation structure and Plant community distribution along environmental gradients of Seqela Dry Afromontane Forest in Northwestern Ethiopia

PLOS ONE 

Article type: Research Paper

We would like to thank the reviewers for their efforts in reviewing the manuscript and providing valuable suggestions for improving the article. All comments and suggestions were addressed. The changes are marked in yellow color as shown in the revised manuscript. We detailed response for each comment is provided below. 

Editor 

The authors have appropriately revised the manuscript and answered all the questions of the reviewers satisfactorily. I recommend the revised version of the manuscript be accepted for publication in PLOS ONE after minor revision.

Author’s Response: Thanks a lot for letting us revise our paper. Below are the responses to a reviewer one comments.

Reviewer #1: 

1. General comments 

The authors have satisfactorily addressed the majority of my concerns and comments, although some minor ones still need to be addressed.

 Author’s Response: Thanks a lot for this very positive feedback.

2. Title: It needs minor revision “Woody species diversity, structure, and community distribution along environmental gradients of Seqela Dry Afromontane Forest in Northwestern Ethiopia”

 Author’s Response: Thank you for this.

3. In paragraph 2, replace “[25; 26]” with “[25, 26]. 

“… environmental factors [e.g. 27- 31]” please restate this statement. Additionally, use proper punctuation and writing techniques throughout your paper such as, replace “27- 31” with “27–31”. 

Author’s Response: The correction has been made in the revised manuscript.

4. Study area

Paragraph 2 replaces “Eragrostis teff” with “Eragrostis tef”.

Author’s Response: Thank you very much for your excellent observation. We have corrected in the revised version of the manuscript.

Data Analysis

Plant community classification 

##There is a mix of information herein: “agglomerative hierarchical cluster analysis”? Select either “agglomerative cluster analysis” or “hierarchical cluster analysis” with acceptable reason.

##The author did not give sufficient response for this question raised in the first draft.

Author’s Response: Based on your previous comment, we have corrected it as “agglomerative hierarchical cluster analysis”. Let's clarify this comment.

Cluster analysis in ecology is a statistical method used to classify objects (such as species, communities, or individuals) into groups or clusters based on similarity. There are several types of cluster analysis techniques commonly used in ecology. Basically the methods of cluster analysis fall into hierarchical and non-hierarchical (partitioning) techniques.

1. Hierarchical clustering 

Hierarchical clustering is an assembling technique of vegetation stands that could be classified into groups, the processes being repeated at different levels to form a tree.

There are two types of hierarchical clustering methods, namely agglomerative and divisive. 

A. Agglomerative hierarchical clustering: It starts with each object as a separate cluster and then iteratively merges the closest clusters until only one cluster remains. called “bottom up method”, construct a tree-like hierarchy,

B. Divisive Hierarchical Clustering : Divisive hierarchical clustering method is the opposite of agglomerative classification in that the clustering start at the top and work downwards, beginning with the whole collection of plots, and dividing them to form successive subclasses and ultimately form a more or less similar groups. This method is not too common method for plant classification. 

• Agglomerative Hierarchical Clustering is often preferred for plant community classification because it starts with individual data points and progressively groups them together based on similarity, mirroring the natural progression of plant community formation where individual plants come together to form larger groupings. This approach aligns well with the hierarchical structure often observed in plant communities, making it a suitable method for classification.

2. Partitioning (non-hierarchical clustering): Partitioning is a method of classifying vegetation stands in the same level, not in hierarchy ( eg. K- means clustering, Fuzzy clustering).

Results 

Plant community types

## “Figure 3. Dendrogram”, here is my big question why not 2 types or 3 community types or else? Because the author did not show the value of the cophenetic correlation coefficient where they determined the cutree. The author did not give sufficient response for this question raised in the first round of review rather they replied very subjective suggestions.

Author’s Response: Again, Let's clarify this comment. The cophenetic correlation coefficient can be used as some kind of measure of the goodness of fit of a particular dendrogram. Based on the agglomerative hierarchical cluster analysis performed in the study area, four plant community types were identified. The cophenetic correlation coefficient reached a value of 0.774, indicating a considerable degree of similarity in the classification. This information included in the revised manuscript.

Figure 3. Dendrogram”, here is my big question why not 2 types or 3 community types or else?

Author’s Response: we appreciate you raising this question. I (corresponding author) frequently inquire about this question when examining the students or reviewing manuscripts. Determining the number of groups in a cluster analysis is often the primary goal. Among the methods we used to decide the optimum cluster is Elbo method, finally we have got 4 plant community types.

---

## [Decision Letter · Decision Letter 2]

19 Jun 2024

PONE-D-24-05897R2Woody species diversity,structure and community distribution along environmental gradientsof Seqela Dry Afromontane Forest in Northwestern EthiopiaPLOS ONE

Dear Dr. Birhanu,

Thank you for submitting your manuscript to PLOS ONE. After careful consideration, we feel that it has merit but does not fully meet PLOS ONE’s publication criteria as it currently stands. Therefore, we invite you to submit a revised version of the manuscript that addresses the points raised during the review process.

**ACADEMIC EDITOR:**:I recommend the revised version of the manuscript be accepted for publication in PLOS One after minor revision.==============================

We look forward to receiving your revised manuscript.

Kind regards,

Faham Khamesipour, Ph.D.

Academic Editor

PLOS ONE

Journal Requirements:

Reviewers' comments:

Reviewer's Responses to Questions

**Comments to the Author**

1. If the authors have adequately addressed your comments raised in a previous round of review and you feel that this manuscript is now acceptable for publication, you may indicate that here to bypass the “Comments to the Author” section, enter your conflict of interest statement in the “Confidential to Editor” section, and submit your "Accept" recommendation.

Reviewer #1: All comments have been addressed

Reviewer #3: (No Response)

2. Is the manuscript technically sound, and do the data support the conclusions?

Reviewer #1: Yes

Reviewer #3: Partly

3. Has the statistical analysis been performed appropriately and rigorously? 

Reviewer #1: Yes

Reviewer #3: No

4. Have the authors made all data underlying the findings in their manuscript fully available?

Reviewer #1: Yes

Reviewer #3: No

5. Is the manuscript presented in an intelligible fashion and written in standard English?

Reviewer #1: Yes

Reviewer #3: No

6. Review Comments to the Author

Reviewer #1: I found that the authors have addressed the comments I made in the second round revised manuscript. I appreciate the great efforts that the authors have made in response to my questions and concerns.

Thus, I recommend the manuscript for publication after a correction of some minor technical writing problems such as, Introduction-line 7: replace "10-12" with "10–12"; line 10: replace "15, 16" with "15,16" and the like.

Reviewer #3: The study focuses on an interesting and timely area of research. However, the write up, organization, grammar and especially the methodology are not to the standard. The methodology missed how the environmental factors such as altitude, aspect and slope were considered during plot establishment, while most of the data and the conclusion relied on. There are some data such as Shannon diversity index and evenness already collected, but not presented.

7. PLOS authors have the option to publish the peer review history of their article (what does this mean?). If published, this will include your full peer review and any attached files.

Reviewer #1: No

Reviewer #3: No

---

## [Author Response · Author response to Decision Letter 2]

5 Aug 2024

We have corrected the 3rd revised version of the manuscript.

---

## [Decision Letter · Decision Letter 3]

25 Sep 2024

PONE-D-24-05897R3Woody species diversity,structure and community distribution along environmental gradientsof Seqela Dry Afromontane Forest in Northwestern EthiopiaPLOS ONE

Dear Dr. Birhanu,

Thank you for submitting your manuscript to PLOS ONE. After careful consideration, we feel that it has merit but does not fully meet PLOS ONE’s publication criteria as it currently stands. Therefore, we invite you to submit a revised version of the manuscript that addresses the points raised during the review process.

**ACADEMIC EDITOR: **The manuscript is ready for publication after minor revision. It requires some minor improvements for enhanced clarity.==============================

We look forward to receiving your revised manuscript.

Kind regards,

Faham Khamesipour, Ph.D.

Academic Editor

PLOS ONE

Journal Requirements:

Reviewers' comments:

Reviewer's Responses to Questions

**Comments to the Author**

1. If the authors have adequately addressed your comments raised in a previous round of review and you feel that this manuscript is now acceptable for publication, you may indicate that here to bypass the “Comments to the Author” section, enter your conflict of interest statement in the “Confidential to Editor” section, and submit your "Accept" recommendation.

Reviewer #3: All comments have been addressed

Reviewer #4: All comments have been addressed

2. Is the manuscript technically sound, and do the data support the conclusions?

Reviewer #3: Yes

Reviewer #4: Yes

3. Has the statistical analysis been performed appropriately and rigorously? 

Reviewer #3: Yes

Reviewer #4: Yes

4. Have the authors made all data underlying the findings in their manuscript fully available?

Reviewer #3: Yes

Reviewer #4: Yes

5. Is the manuscript presented in an intelligible fashion and written in standard English?

Reviewer #3: No

Reviewer #4: Yes

6. Review Comments to the Author

Reviewer #3: - The authors have addressed the comments given in the first round revision and the manuscript is very much improved. The comments for further improvement include

1. The size of the study forest is not mentioned. However, in the discussion under woody species composition, a sentence

saying the variation in Shannon diversity index might be due to the size of the forest is included. Therefore, including the

size the forest in the paper helps to see such comparisons

2. I couldn't see disturbance data in this manuscript. However, a significant portion of the conclusion is about the effect of

disturbances. Thus, the conclusion should focus on the findings of the manuscript.

3. The width of the bars in Figure 3 is not to the standard

Reviewer #4: Please correct the species names in the abstract. The objectives can be fine tuned to align with the results.

7. PLOS authors have the option to publish the peer review history of their article (what does this mean?). If published, this will include your full peer review and any attached files.

Reviewer #3: No

Reviewer #4: **Yes: **Emiru Birhane

---

## [Author Response · Author response to Decision Letter 3]

2 Oct 2024

We would like to thank the reviewers for their efforts in reviewing the manuscript and providing valuable suggestions for improving the article. All comments and suggestions were addressed. The changes are marked in yellow color as shown in the revised manuscript. We detailed response for each comment is provided below. 

Reviewer #3: - The authors have addressed the comments given in the first round revision and the manuscript is very much improved. The comments for further improvement include

1. The size of the study forest is not mentioned. However, in the discussion under woody species composition, a sentence saying the variation in Shannon diversity index might be due to the size of the forest is included. Therefore, including the size the forest in the paper helps to see such comparisons

2. I couldn't see disturbance data in this manuscript. However, a significant portion of the conclusion is about the effect of

disturbances. Thus, the conclusion should focus on the findings of the manuscript.

3. The width of the bars in Figure 3 is not to the standard

Author’s Response: Thank you for your valuable feedback; we have included the size of the Seqela Forest (197.6 ha) in the manuscript, revised the conclusion to focus on the findings, and corrected the width of the bars in Figure 3 to meet standard requirements.

Reviewer #4: Please correct the species names in the abstract. The objectives can be fine tuned to align with the results.

Author’s Response: Thank you for your valuable feedback. We have corrected the objectives in the revised manuscript to align more closely with the results. However, we would like to clarify that the species names in the abstract are correct as originally presented, except for Acacia abyssinica, which has been updated to Vachellia abyssinica as per the recent taxonomic classification.

---

## [Decision Letter · Decision Letter 4]

17 Oct 2024

Woody Species Diversity, Structure and Community Distribution Along Environmental Gradients of Seqela Dry Afromontane Forest in Northwestern Ethiopia

PONE-D-24-05897R4

Dear Dr. Birhanu,

We’re pleased to inform you that your manuscript has been judged scientifically suitable for publication and will be formally accepted for publication once it meets all outstanding technical requirements.

Kind regards,

Faham Khamesipour, Ph.D.

Academic Editor

PLOS ONE

Additional Editor Comments (optional):

Reviewers' comments:

Reviewer's Responses to Questions

**Comments to the Author**

1. If the authors have adequately addressed your comments raised in a previous round of review and you feel that this manuscript is now acceptable for publication, you may indicate that here to bypass the “Comments to the Author” section, enter your conflict of interest statement in the “Confidential to Editor” section, and submit your "Accept" recommendation.

Reviewer #3: All comments have been addressed

Reviewer #4: All comments have been addressed

2. Is the manuscript technically sound, and do the data support the conclusions?

Reviewer #3: Yes

Reviewer #4: Yes

3. Has the statistical analysis been performed appropriately and rigorously? 

Reviewer #3: Yes

Reviewer #4: Yes

4. Have the authors made all data underlying the findings in their manuscript fully available?

Reviewer #3: Yes

Reviewer #4: Yes

5. Is the manuscript presented in an intelligible fashion and written in standard English?

Reviewer #3: Yes

Reviewer #4: Yes

6. Review Comments to the Author

Reviewer #3: The comments given are well entertained and the manuscript can be published in the present condition

Reviewer #4: The comments are addressed. No further comment. The authors critically addressed the comments given to improve the paper.

7. PLOS authors have the option to publish the peer review history of their article (what does this mean?). If published, this will include your full peer review and any attached files.

Reviewer #3: No

Reviewer #4: **Yes: **Emiru Birhane

---

## [Editor Report · Acceptance letter]

19 Nov 2024

PONE-D-24-05897R4 

PLOS ONE

Dear Dr. Birhanu, 

I'm pleased to inform you that your manuscript has been deemed suitable for publication in PLOS ONE. Congratulations! Your manuscript is now being handed over to our production team.

Kind regards, 

on behalf of

Dr. Faham Khamesipour 

Academic Editor

PLOS ONE